# Effects of Dietary Chenodeoxycholic Acid Supplementation in a Low Fishmeal Diet Containing *Clostridium autoethanogenum* Protein on Growth, Lipid and Cholesterol Metabolism, and Hepatopancreas Health of *Litopenaeus vannamei*

**DOI:** 10.3390/ani13132109

**Published:** 2023-06-26

**Authors:** Menglin Shi, Chaozhong Zheng, Yidan Sun, Xiaoyue Li, Guilun He, Junming Cao, Beiping Tan, Shiwei Xie

**Affiliations:** 1Laboratory of Aquatic Nutrition and Feed, College of Fisheries, Guangdong Ocean University, Zhanjiang 524088, China; shimenglin0301@163.com (M.S.); realczzheng@163.com (C.Z.); 18865697780@163.com (Y.S.); lxy303587@163.com (X.L.); 15018045717@163.com (G.H.); junmcao@163.com (J.C.); bptan@126.com (B.T.); 2Aquatic Animals Precision Nutrition and High-Efficiency Feed Engineering Research Centre of Guangdong Province, Zhanjiang 524088, China; 3Key Laboratory of Aquatic, Livestock and Poultry Feed Science and Technology in South China, Ministry of Agriculture, Zhanjiang 524088, China; 4Guangdong Provincial Key Lab of Aquatic Animals Disease Control and Healthy Culture, Zhanjiang 524088, China

**Keywords:** CDCA, lipid metabolism, cholesterol metabolism, intestinal immunity, *Litopenaeus vannamei*

## Abstract

**Simple Summary:**

Due to the scarcity of fishmeal resources, alternatives that provide the necessary nutrients for *Litopenaeus vannamei* are being sought. While fishmeal has been the main nutrition source of *Litopenaeus vannamei* due to its rich nutrients, the search for protein sources to replace it is becoming increasingly important. The protein derived from *Clostridium autoethanogenum* protein shows great potential as a substitute for fishmeal owing to its exceptional nutritional value. However, previous studies have suggested that the growth of *Litopenaeus vannamei* may be adversely affected by the fishmeal replacement of *Clostridium autoethanogenum* protein. Research has demonstrated that chenodeoxycholic acid can improve the growth and intestinal health of *Litopenaeus vannamei*. Five diets were formulated with varying concentrations of chenodeoxycholic acid to feed *Litopenaeus vannamei*. The results revealed that replacing partial fishmeal with *Clostridium autoethanogenum* protein and adding 0.06% chenodeoxycholic acid led to significant improvements in the growth performance, lipid metabolism, and sterol metabolism. Additionally, the hepatopancreas health of the *Litopenaeus vannamei* was also improved. The conducted experiment has set the groundwork for further investigation into determining the most effective quantity of chenodeoxycholic acid to be utilized in low fishmeal diets.

**Abstract:**

The study aimed to assess the impact of adding chenodeoxycholic acid (CDCA) to the diet of *Litopenaeus vannamei* on their growth performance, lipid and cholesterol metabolism, and hepatopancreas health while being fed a low fishmeal diet. Five diets were formulated, one of which contained 25% fishmeal (PC); fishmeal was partially replaced with *Clostridium autoethanogenum* protein in the remaining four diets and supplemented with 0, 0.03, 0.06, and 0.09% CDCA (NC, BA1, BA2, and BA3, respectively). In this study, four replicates of each diet were assigned and each replicate consisted of 30 shrimp with an average weight of (0.25 ± 0.03 g). The shrimp were fed four times a day for a period of 56 days. The results of this study indicate that the inclusion of CDCA in the diet had a positive impact on the growth performance of the shrimp. The final body weight (FBW), weight gain (WG), and specific growth rate (SGR) of the shrimp in the PC group were similar to those in the BA2 group, and significantly higher than those in the other three groups. The survival rate (SR) was similar among all groups. In comparison to the PC group, the low fishmeal groups exhibited a significant decrease in the crude lipid content of the whole shrimp, as well as the Total cholesterol (T-CHOL), Low-density lipoprotein (LDL-C), and High-density lipoprotein (HDL-C) levels in the hemolymph. Regarding the sterol metabolism, the dietary supplementation of CDCA up-regulated the mRNA expression of intracellular cholesterol transporter 1-like (*npc1*), 7-dehydrocholesterol reductase (*7dhcr*), Delta (24) sterol reductase (*Δ24*), HMG-CoA reductase membrane form (*hmgcr*), and sterol carrier protein 2 (*scp*). In the lipid metabolism, the mRNA expression of sterol-regulatory element binding protein (*srebp*) was significantly down-regulated in the shrimp fed the BA1 diet and the expression of AMP-activated protein kinase (*ampk*) was significantly up-regulated in the shrimp fed the BA1 and BA3 diets compared to the PC group. The mRNA expression of triacylglycerol lipase (*tgl*) was significantly up-regulated in the shrimp fed the BA2 diet compared to the NC group. Compared with the shrimp fed the PC diets, the dietary supplementation of CDCA significantly down-regulated the protein expression of SREBP1. The lumen damage in the BA1 group was significantly less severe than those in the NC group. The addition of 0.06% CDCA to low fishmeal diets can improve the growth performance, lipid and cholesterol metabolism, and hepatopancreas health of *L. vannamei*.

## 1. Introduction

At present, *L. vannamei* a highly sought after aquatic product globally due to its abundance of high-quality protein [1,2,3]. The special palatability, high nutritional value, and protein content of fishmeal make it the primary protein source of aquatic organisms [4]. According to the current demand for fishmeal used in aquatic feed, the price of fishmeal is likely to go up by 30% in 2030 [5]. Thus, it is pressing to find alternative protein sources for fishmeal.

Previous studies have confirmed that concentrated cotton meal protein [6], soy protein concentrate [7], soybean meal [8], high-protein cottonseed meal [9], corn protein concentrate [10], methanotroph [11], black soldier fly [12], and yeast cultures [13] can be used as partial substitutes for fishmeal. Plant proteins are the main alternative to fishmeal due to their high protein yield and low cost [14]. Despite being a cost-effective and high-yielding alternative to fishmeal, plant proteins are not without their limitations. One of the major drawbacks of plant proteins is their high content of anti-nutritional factors, which can negatively impact their nutritional value. Additionally, plant proteins are often less palatable than fishmeal, which can affect their overall acceptability as a feed ingredient [15]. An increasing number of studies are proving that some plant proteins can hurt the intestine of shrimp [1,16]. Bacterial proteins do not have the same disadvantages as plant proteins and have been used, with good results, as an alternative to fishmeal. The advantages of bacterial proteins are their high nutritional value and easy cultivation [17]. As a result, increasing studies are focusing on bacterial proteins. *Clostridium autoethanogenum* proteins (CAP) are a novel type of bacterial protein, which is a by-product of the fermentation and combustion of ethanol in the steelmaking industry. Carbon monoxide and ammonia are the main raw materials for CAP production [18]. Compared with plant proteins, there is a higher crude protein component and lower anti-nutritional factors in CAP [18]. Previous studies have demonstrated that replacing 30% of fishmeal with CAP does not adversely affect the growth and muscle quality of *L. vannamei* [18]. Similarly, other studies have shown that CAP substituting 40% fishmeal did not adversely affect *L. vannamei* [19,20]. Moreover, the partial substitution of fishmeal with CAP has no adverse effects on yellow croaker (*Larimichthys crocea*) [21] and largemouth bass (*Micropterus salmoides*) [22].

However, the growth of aquatic animals would be impacted if an excessive quantity of fishmeal was replaced by CAP. When CAP replaced fishmeal by more than 30%, the growth, flesh quality, and immune function of *L. vannamei* were harmed to varying degrees [17,23]. Replacing 40% of fishmeal with CAP also had an effect on the lipid metabolism of *L. vannamei*, and excessive CAP may have negative effects on largemouth bass (*Micropterus salmoides*) [24,25]. Dietary supplementation with functional feed additives can reduce the adverse effects on shrimp. Chenodeoxycholic acid (CDCA) is a primary bile acid, which can also form secondary bile acids (lithocholic acid, deoxycholic acid, ursodeoxycholic acid) with glycine or taurine for the intestinal–hepatic cycle. Intestinal L-cells stimulate the secretion of cholecystokinins, which are then released from the gallbladder into the intestine via the common bile duct to aid in the absorption of lipids and fat-soluble vitamins [1]. Bile acids are catabolic products of cholesterol, which is then secreted by the bile ducts and digested by the intestines to be absorbed as a fat-soluble substance [26]. Due to the amphoteric structure of bile acids, they are essential for promoting the intestinal metabolism, as well as the further digestion of fat-soluble vitamins, cholesterol, and lipids [27,28]. According to earlier research, supplementing bile acids can enhance the growth performance and lipid metabolism of *L. vannamei* [27]. In addition, many studies have indicated that bile acids can enhance immune function and regulate the intestinal health of aquatic animals [29]. Early research has demonstrated that CDCA can boost the immune response and improve intestinal health in aquatic animals [30]. Prior studies have demonstrated that the addition of CDCA to plant protein feed can enhance the growth performance and intestinal health of shrimp [1,14]. However, limited research has been conducted on the impact of CDCA supplementation in low fishmeal diets on the lipid metabolism and immune response of *L. vannamei*.

The aim of this study was to examine the effects of adding CDCA to a low fishmeal diet on the growth performance, body composition, lipid metabolism, and intestinal health of *L. vannamei*.

## 2. Materials and Methods

### 2.1. Experimental Diets

In this study, five iso-nitrogenous and iso-lipid diets were formulated, including a diet containing 25% fishmeal (PC) and four diets containing 12.5% fishmeal and supplemented with 0, 0.03, 0.06, and 0.09% CDCA (NC, BA1, BA2, and BA3, respectively). Fishmeal was substituted with 10% CAP and supplemented with fish oil. Table 1 displays the proximate composition analyses and formulation details of the diets utilized in this study.

### 2.2. Shrimp and Experimental Conditions

Shrimp were purchased from the Guangdong Yuehai Seed Co., Ltd. (Zhanjiang, China). The shrimp were given commercial feed (48.0% crude protein and 8.0% crude lipid) for one week prior to the experiment. After 24 h of starvation, 600 healthy shrimp (0.25 ± 0.03 g) were selected and randomly allocated into 5 groups. Each group consisted of 4 replicates, with each replicate containing 30 shrimp. They were fed at 07:15, 12:15, 17:15, and 22:15 every day for 56 days. Water temperatures were 27–30 °C, the pH was 7.7–8.1, and the salinities were around 31‰.

### 2.3. Sampling and Chemical Analysis

Following a feeding period of 56 days, the shrimp in each tank were subjected to 24 h fasting. The weight of the shrimp was measured and the survival rate was determined through calculations. To determine the moisture content, crude lipid, and crude protein levels, a random sample of five shrimp was chosen from a tank. The hemolymph of five shrimp per tank was collected and subsequently centrifuged at 4 °C and 3500 rpm for 15 min. After the hemolymph supernatant was obtained, it was carefully collected and immediately preserved at a temperature of −80 °C. Hepatopancreas of four shrimp were taken, washed in normal saline (0.9% NaCl solution), then transferred to liquid nitrogen, and then stored at −80 °C for subsequent analysis. The intestine and hepatopancreas were collected from four shrimp, kept in RNA later reagent (Ambion^®^, ThermoFisher, Waltham, MA, USA), and stored at −80 °C.

The whole shrimps underwent a drying process in a 105 °C oven, which was carried out with the aim of assessing their moisture content. The determination of the crude protein content was carried out using the Dumas Nitrogen method with a Primacs100 analyzer (Skalar, Breda, Dutch). For the determination of the crude lipid, the XT15 extractor (Ankom, NY, USA) was utilized. Additionally, the amino acid content was analyzed in accordance with the standard GB/T 18246-2019.

### 2.4. Enzyme Activity Assays

The contents of triglyceride (TG), Total cholesterol (T-CHOL), High-density lipoprotein (HDL-C), Low-density lipoprotein (LDL-C), and the activities of alanine aminotransferase (ALT) and aspartate aminotransferase (AST) in the hemolymph were analyzed using the commercial Kit of Nanjing Jiancheng (Nanjing Jiancheng Institute Biological Engineering, China). The total cholesterol (T-CHOL) content and the activity of lipase in the hepatopancreas were determined according to the commercial Kit of Nanjing Jiancheng (Nanjing Jiancheng Institute Biological Engineering, Nanjing, China).

### 2.5. Quantitative Real-Time PCR Analysis

Total RNA was extracted from the hepatopancreas and intestine according to Trizol Reagent (Transgen Biotech, Beijing, China). The extracted RNA was reverse transcribed into cDNA using the QuantiTect Reverse Transcription Kit (Accurate Biotechnology, Changsha, China). The primers used in the experiment were synthesized by the Sangon Biotech (Guangdong, China) Co., Ltd. The sequence is shown in Table 2. Subsequently, real-time fluorescence quantitative PCR was performed using Light Cycler 480 (Roche Applied Science) to detect gene expression.

### 2.6. Morphological Analysis of Hepatopancreas

Hepatopancreas samples were soaked in 4% formaldehyde for 24 h before being transferred to Bonn solution and 70% ethanol. After that, the samples were dehydrated with different concentrations of ethanol, followed by embedding in paraffin. The sections were stained with hematoxylin-eosin and observed under microscope (Nikon Ni-U, Tokyo, Japan).

### 2.7. Western Blot Analyses

According to the method of Xu et al. [31], 30 mg of hepatopancreatic tissue was dissected, PBS, cell lysate, protease inhibitor, and phosphorylase inhibitor were added, and finally, PMSF was added for disruption. The concentration of hepatopancreatic protein was measured using the BCA method and subsequently adjusted to a concentration of 4 mg/mL with PBS and loading buffer. Afterwards, the protein samples underwent denaturation in boiling water for a duration of 10 min before being utilized for the separation of proteins (36 mg of total protein per gel hole) through SDS-PAGE. After electrophoresis, they were transferred to membrane and block with rapid blocking solution for 25 min at room temperature. After development with ECL reagents, bands were quantified using Image J (version 1.43, National Institutes of Health). In this study, the following antibodies were used: antibodies against SREBP1(1:800, ab28481, Abcam), FXR (1:1500, bs-12867R, Bioss), GAPDH (1:1000, 2118 S, Cell Signaling Technology).

### 2.8. Calculations and Statistical Analysis

The parameters were calculated as follows:Weight gain (WG, %) = 100 × (m1 − m0)/m0
Specific growth rate (SGR, % day − 1) = 100 × (Ln m1 − Ln m0)/t
Survival rate (SR, %) = 100 × (n1)/(n0)
Feed conversion rate (FCR) = feed consumed (g)/(m1 − m0)
Feeding intake (FI) = feed consumed/n1
Protein efficiency rate (PER) = (m1 − m0)/protein intake
where m0, initial body weight; m1, final body weight; n0, initial amount of shrimp; n1, final amount of shrimp.

Prior to conducting a one-way analysis of variance (ANOVA), the data underwent normality distribution testing using the Kolmogorov-Smirnov test and homogeneity of variances testing using Levene’s test. The data were then analyzed using one-way ANOVA followed by Tukey’s multiple range test to identify significant differences among treatments. The statistical software used was SPSS 21.0 (SPSS, Chicago, IL, USA). A probability value of *p* < 0.05 was considered statistically significant.

## 3. Results

### 3.1. Growth Performance

Based on the data presented in Table 3, the final body weight (FBW), weight gain (WG), and specific growth rate (SGR) of the shrimp in the PC group were similar to those in the BA2 group, and significantly higher than those in the other three groups (*p* < 0.05). The survival rate (SR) was similar among all groups (*p* > 0.05). The feed conversion rate (FCR) and feeding (FI) of the five experimental groups had no differences (*p* > 0.05). The protein efficiency rate (PER) of the shrimp in the PC and BA1 groups was significantly higher than that of the NC group (*p* < 0.05).

### 3.2. Proximate Composition of Whole Shrimp

As indicated in Table 4, no noticeable differences in the moisture content were observed among the five groups. (*p* > 0.05). The NC group exhibited a significantly lower crude protein content in comparison to the BA1 and BA3 groups (*p* < 0.05). The crude lipid content of the shrimp was significantly decreased when CDCA was added into their diet (*p* < 0.05).

### 3.3. Hemolymph and Hepatopancreas Biochemical Indicators

According to Table 5, there was no significant difference in the content of TG in the hemolymph (*p* > 0.05). The T-CHOL content in the PC group was significantly higher than the other four groups (*p* < 0.05). The HDL-C content of the shrimp fed the BA2 diet was significantly higher than those fed the NC and BA1 diet (*p* < 0.05). The LDL-C and ALT content significantly increased with the increase in the CDCA content (*p* < 0.05). The AST content of the shrimp fed the BA3 diet was significantly higher than those fed the NC diet (*p* < 0.05). The hepatopancreas T-CHOL content in the shrimp fed the PC diet was significantly higher than those fed the other diets, except for the BA3 diet (*p* < 0.05). The lipase content in the hepatopancreas was significantly lower in the PC diet than that in the BA1 diet (*p* < 0.05).

### 3.4. Lipid and Sterol Metabolism-Related Genes Expression

The relative expression of lipid metabolism-related genes in the hepatopancreas is shown in Figure 1. Among the genes related to lipid metabolism, the mRNA expression of tgl in the BA2 group was significantly higher than that in the NC group, and the mRNA expression of ampk in the BA1 and BA3 groups were significantly higher than that in the PC group (*p* < 0.05). Compared with the shrimp fed the PC diet, the supplementation of CDCA in the diets down-regulated the mRNA expression of srebp (*p* < 0.05).

As shown in Figure 1, adding CDCA to low fishmeal diets affected the expression levels of sterol metabolism-related genes in the hepatopancreas of *L. vannamei*. Compared with the shrimp fed the NC diets, the dietary supplementation of CDCA up-regulated the mRNA expression of *npc1*, *7dhcr*, and *Δ24* (*p* < 0.05). The mRNA expression of *hmgcr* significantly increased in the BA2 and BA3 diets compared to the NC diet (*p* < 0.05). In all groups, the mRNA expression of Steroid reductase (*sr*) in the hepatopancreas was similar (*p* > 0.05). The shrimp fed the BA2 diet achieved the highest scp expressions compared to the other groups (*p* < 0.05).

### 3.5. Phenol Oxidase-Related Genes Expression

The relative expression of phenol oxidase-related genes in the hepatopancreas is shown in Figure 2. Among the genes related to the phenol oxidase system, the mRNA expression of prophenoloxidas (proPO) and lipopolysaccharide and beta-1,3-glucan binding protein (lgbp) in the NC group was significantly higher than that in the PC group; the mRNA expression of proPO, lgbp, and prophenoloxidase activating factor (ppaf) showed a decreasing and then increasing trend with of the increase in CDCA levels (*p* < 0.05).

### 3.6. Morphological Analysis of Hepatopancreas

The morphology of the hepatopancreas is shown in Figure 3. The hepatopancreas of *L. vannamei* is composed of multiple hepatic tubules. The large vacuoles in the hepatic tubules are B cells (Blasenzellen cells), the small lipid vacuoles are R cells (Restzellen cells), and the area with a five- or four-pointed star is the lumen of the hepatopancreatic tubules (Lumen of hepatopancreatic tubules). The HE-stained hepatopancreas showed that with the increase in CDCA addition, the number of B cells in the hepatic tubule gradually increased, and the number of R cells gradually decreased. The lumen of the shrimp in the NC group was dilated or even ruptured. The lumen of the shrimp in the BA1 group was slightly damaged, and the lumen structure of the shrimp in the other groups was normal.

### 3.7. Western Blot Analysis of Bile Acid Metabolism Related Proteins Expression

As shown in Figure 4, compared with the shrimp fed the PC diet, the dietary supplementation of 0.06% CDCA significantly down-regulated the protein expression of SREBP1 (*p* < 0.05). In all of the groups, the protein expression of FXR in the hepatopancreas was similar (*p* > 0.05).

## 4. Discussion

Although many studies have proven that excessive dietary CAP replacement for fishmeal would cause some adverse effects, this study revealed that substituting partial fishmeal with CAP could still significantly enhance the growth performance of *L. vannamei* by adding appropriate CDCA. The addition of 0.06% CDCA can effectively mitigate this negative effect. The study revealed that the partial replacement of fishmeal with CAP significantly reduced the growth of *L. vannamei* [17]. According to previous research, the growth of *L. vannamei* significantly decreased when fishmeal was substituted with more than 30% CAP [23].

Similar to our results, a low fishmeal diet supplemented with CDCA can improve the growth performance and intestinal health of *L. vannamei* [32], and dietary supplementation of CDCA can also improve the hepatopancreas health of *L. vannamei* [1]. The growth of *L. vannamei* can be stimulated through dietary supplementation of 0.02–0.03% bile acid, as reported by Su et al. [27]. Iwashita et al. [33] demonstrated that supplementing plant protein-based feeds with choline taurine (a type of bile acid) resulted in the improved growth of rainbow trout (*Oncorhynchus mykiss*). Adding CDCA to feeds can enhance the growth performance of shrimps, probably due to their ability to promote the emulsification and digestion of feeds [34]. In this study, the growth performance of shrimp was not further improved with the increase in the bile acid content, indicating that the addition of excessive CDCA will cause negative effects. This is similar to the findings of Su et al. [24]. In fact, the excessive addition of CDCA had a negative impact on shrimp growth. Jiang et al. [35] found that a whole plant protein diet supplemented with 0.015–0.045% bile acid significantly improved the growth of gefilte fish (*Oreochromis niloticus*), but when the level of bile acid addition was increased to 0.135%, it hindered the growth. In terms of feed utilization, the FCR and FI were similar among all the groups. The addition of 0.03% CDCA to low fishmeal feeds can significantly improve protein efficiency. Studies conducted previously have shown that the addition of 0.06% bile acid to the diet of grass carp (*Ctenopharyngodon idella*) can mitigate the reduction in protein efficiency associated with plant-based raw materials [36].

Adding CDCA at a concentration range of 0.03%~0.09% has been found to enhance the crude protein content of whole shrimp. Another study reported that adding 0.006% bile acid significantly improved the whole fish crude protein of grass carp, and it was speculated that the muscle glutamate dehydrogenase involved in protein catabolism was downregulated by adding bile acid, thus promoting the protein synthesis required for somatic cell growth [36]. This experiment has also shown that either replacing fishmeal with CAP or adding CDCA to fishmeal could significantly reduce the crude lipid composition of whole shrimp. Studies of grass carp [35,37] showed that a low dose of dietary bile acid could reduce the accumulation of body lipid. This may be due to the enhancement of triglycerides breakdown promoting the lipids’ decomposition. Bile acid is an essential component in the process of lipid emulsification as it plays a critical role in breaking down large lipid molecules into smaller ones, which in turn increases the surface area for lipase reaction, which is crucial for the digestion and absorption of lipids [38]. Both the replacement of fishmeal with CAP and the supplementation of CDCA can play a role in enhancing lipase activity in the hepatopancreas, which corresponds to the trend of crude lipid in whole shrimp. Replacing fishmeal with CAP significantly reduced the content of T-CHOL and HDL-C in the hemolymph, as well as the content of T-CHOL in the hepatopancreas, which are closely related to lipid metabolism [39,40]. However, the supplementation of CDCA could effectively reverse this trend. These findings indicate that the addition of CDCA is crucial for regulating the lipid metabolism in shrimp. The main way in which HDL-C regulates lipid metabolism is through mediating reverse cholesterol transport, which refers to the conversion of cholesterol from the peripheral hepatopancreatic Cholesterol from peripheral tissues, such as lipid and muscle, being taken into the hepatopancreas and converted to bile acids before entering the next level of the metabolic cycle. This process is accompanied by the formation of HDL-C. In addition, Chang Yumei et al. suggested that a decrease in serum T-CHOL may be an indication of liver injury.

The *tgl*—as a regulatory transcription factor for lipolysis—and its translated products can hydrolyze triglycerides to convert lipids into glycerol and free fatty acids [41]. In general, higher *tgl* expression indicates a higher degree of triglyceride hydrolysis. The *ampk* is an important cellular energy sensor, and the activation of the *ampk* stimulates the catabolic pathway of adenosine triphosphate (ATP) breakdown, leading to accelerated fatty acid oxidation and glycolysis, while limiting the production of fatty acids, protein, and glycogen. Polakof et al. [42] showed that the *ampk* limited a reduction in fatty acid synthesis in rainbow trout (*Oncorhynchus mykiss*). In the regulation of lipid synthesis, *srebp* is one of the most important transcription factors [43]. This study found that supplementing shrimp diets with CDCA led to an increase in the expression of the genes related to energy metabolism and lipid oxidation, while inhibiting the genes associated with lipid synthesis, thereby significantly reducing the body lipid in shrimp. The Western blot results showed that dietary supplementation with CDCA significantly reduced SREBP1 expression, indicating reduced lipid synthesis, which was consistent with the molecular results.

Sterol synthesis has an important impact on lipid metabolism [44]. *Hmgrc* plays a crucial role in regulating the biosynthesis of cholesterol in vertebrates by acting as the rate-limiting enzyme in the process [45]. The results of this experiment revealed that a low fishmeal diet dramatically reduced *hmgrc* mRNA expression, but CDCA supplementation increased *hmgrc* expression. The results obtained in this study are consistent with the findings reported by Rozner et al. [46]. *7dhcr* is a critical enzyme in the process of cholesterol synthesis, playing a pivotal role in promoting cholesterol production [47]. In this experiment, low fishmeal diets down-regulated *7dhcr* expression and supplementation with CDCA up-regulated *7dhcr* expression. Sterols are converted to cholesterol through the action of Δ*24sr* and a number of other enzymes [25,48]. Sterols carrier proteins are essential for cholesterol absorption and storage. The addition of CDCA to the diet led to a significant increase in the mRNA expression of *scp* in the BA2 group, implying that it can promote the absorption and utilization of sterols. *Npc1* and *npc2* are crucial proteins involved in intracellular lipid transport. Cholesterol initially binds to *npc2*, and subsequently to *npc1*, which accelerates the transport of cholesterol [49]. In this experiment, the low fishmeal diet down-regulated the expression of *npc1*, while the supplementation of CDCA could up-regulated the expression of *npc1*, which indicated that the addition of CDCA could accelerate intracellular cholesterol transport in shrimp.

ALT and AST are considered to be markers of hepatopancreas damage. In general, the activity of transaminase in the hemolymph is low. However, in cases of hepatopancreatic disease, increased cell membrane permeability can lead to the release of transaminase into the hemolymph [50,51]. According to the findings of this study, the NC diet significantly increased the activities of these two transaminases in the hemolymph, which seemed to indicate that the shrimp hepatopancreas was damaged, and the addition of 0.03–0.09% CDCA could significantly alleviate this damage. According to the study conducted by Su et al. [27], it was observed that the supplementation of bile acid below 0.05% did not affect the hemolymph ALT and AST content of *L. vannamei*, but adding more than 0.05% caused hepatopancreatic toxicity. The difference in these results might be due to the different types of bile acids. On the level of gene expression, the *proPO* and *lgbp* expression levels were significantly increased in the NC, BA2, and BA3 groups. This indicated that the CAP replacement of fishmeal and the supplementation of 0.06–0.09% CDCA have the potential to activate the *proPO* immune system.

In crustaceans, the hepatopancreas are the main organs for absorbing and storing substances, and their morphology indicates the nutritional status of crustaceans [52,53]. The results of Xu et al. [54] showed that R cells are the main site of lipid droplets and glycogen nutrient reserves, and their research also showed that the lower the lipid level or the higher the energy metabolism of the organism, the smaller the number of R cells. B cells are the primary location where the complex process of synthesizing, secreting, and breaking down critical molecules occurs, making them an integral component of the immune system. The more vigorous the synthesis of digestive enzymes, the greater the number of B cells [55]. However, studies have shown that B cell hypertrophy may also be associated with fat oxidative damage [56]. Studies have shown that bile acids were potentially cytotoxic, with previous results confirming that high concentrations (0.135%) of dietary bile acids damaged tilapia liver cells [35,57]. The study’s findings indicate that increased dietary CDCA levels are positively correlated with the number of hepatopancreas B cells, while a negative correlation was observed between the CDCA levels and the number of R cells. The findings indicate that CDCA being added to fishmeal can effectively promote shrimp growth. In addition, the dilated and irregular lumen of the central hepatic tubule is also a sign of hepatopancreas damage. The current research findings demonstrate that the hepatic canalicular lumen of the shrimp in the NC group showed obvious dilatation, but there was no obvious abnormality in the lumen of the shrimp in the BA2 and BA3 groups, indicating that dietary CDCA supplementation can alleviate the hepatopancreas injury of shrimp.

## 5. Conclusions

This study aimed to assess the impact of CDCA supplementation on *L. vannamei* when fed a low fishmeal diet with CAP replacing fishmeal. The study revealed that the addition of 0.06% CDCA could alleviate the negative effects on the growth performance, lipid and cholesterol metabolism, and hepatopancreas health of *L. vannamei*.

## Figures and Tables

**Figure 1 animals-13-02109-f001:**
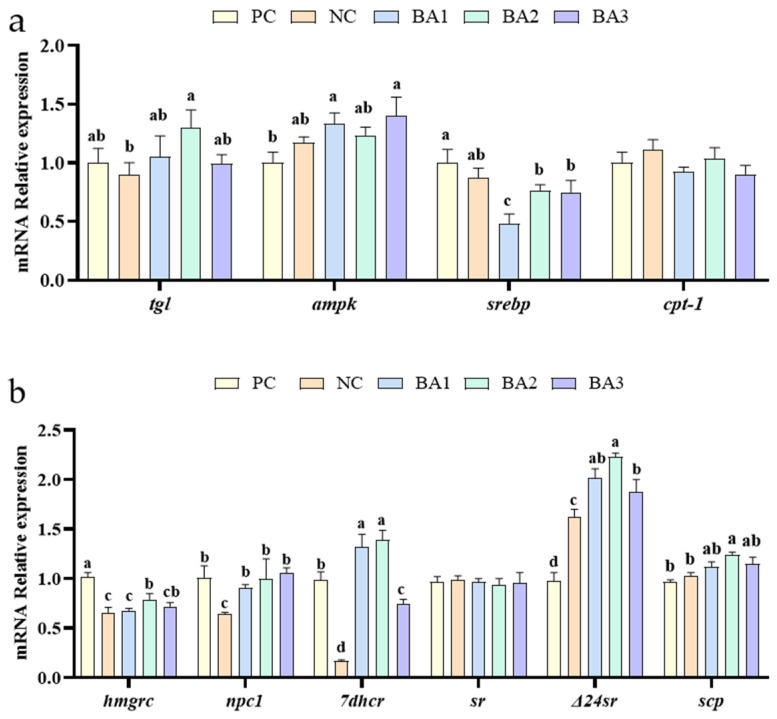
Lipid metabolism and cholesterol metabolism-related gene mRNA relative expression. (**a**) the expression of lipid metabolism-related genes in the hepatopancreas. (**b**) the expression of cholesterol metabolism-related genes in the hepatopancreas. *tgl*: triacylglycerol lipase; *cpt-1*: carnitine palmitoyl transferase-1; *ampk*: AMP-activated protein kinase; sr*ebp*: sterol-regulatory element binding protein; *hmgcr*: HMG-CoA reductase membrane form; *scp*: sterol carrier protein 2; Δ*24sr*: Delta (24) sterol reductase; *sr*: Steroid reductase; *7dhcr*: 7-dehydrocholesterol reductase; *npc1*: NPC1 intracellular cholesterol transporter 1-like. Data represent mean ± SD (*n* = 4). Values in the same row with different letters are significantly different (*p* < 0.05) based on Tukey’s multiple-test. PC, 25% fishmeal; NC, 12.5% fishmeal; BA1, NC + 0.03 g kg^−1^ chenodeoxycholic acid; BA2, NC + 0.06 g kg^−1^ chenodeoxycholic acid; BA3, NC + 0.03 g kg^−1^ chenodeoxycholic acid.

**Figure 2 animals-13-02109-f002:**
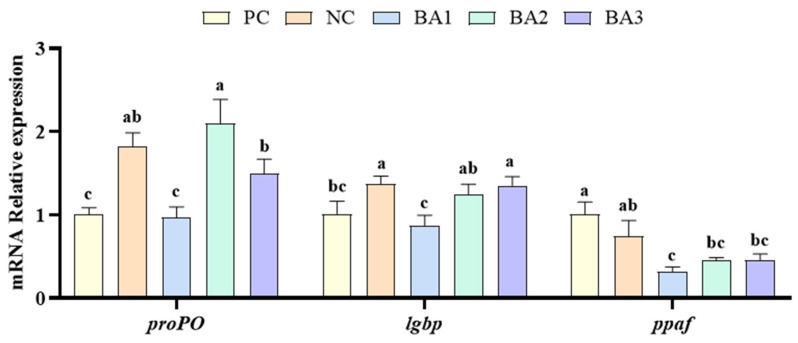
The relative expression of phenol oxidase -related gene. proPO: prophenoloxidas, lgbp: lipopolysaccharide and beta-1,3-glucan binding protein; ppaf: prophenoloxidase activating factor. Data represent mean ± SD (*n* = 4). Values in the same row with different letters are significantly different (*p* < 0.05) based on Tukey’s multiple-test. PC, 25% fishmeal; NC, 12.5% fishmeal; BA1, NC + 0.03 g kg^−1^ chenodeoxycholic acid; BA2, NC + 0.06 g kg^−1^ chenodeoxycholic acid; BA3, NC + 0.03 g kg^−1^ chenodeoxycholic acid.

**Figure 3 animals-13-02109-f003:**
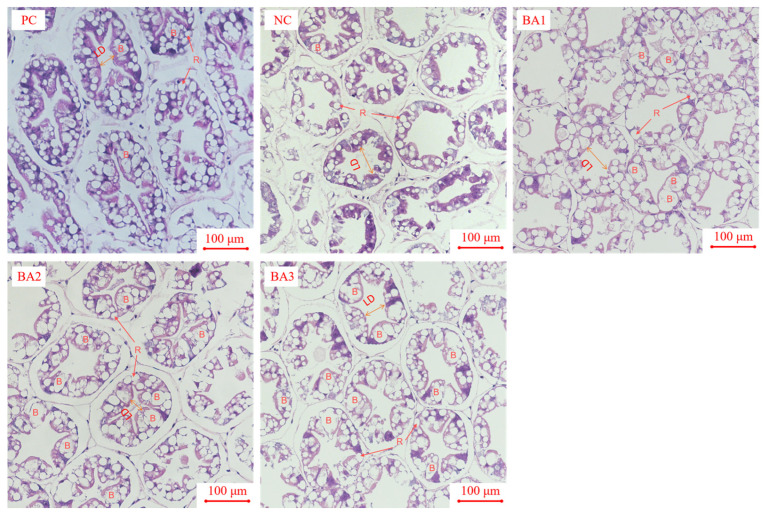
The effects of CDCA addition to low fishmeal diets on hepatopancreas morphology of the shrimp. The letter B indicates B cells (Blasenzellen cells), the letter R indicates R cells (Restzellen cells), and the LDindicates lumen diameter. PC, 25% fishmeal; NC, 12.5% fishmeal; BA1, NC + 0.03 g kg^−1^ chenodeoxycholic acid; BA2, NC + 0.06 g kg^−1^ chenodeoxycholic acid; BA3, NC + 0.00 g kg^−1^ chenodeoxycholic acid.

**Figure 4 animals-13-02109-f004:**
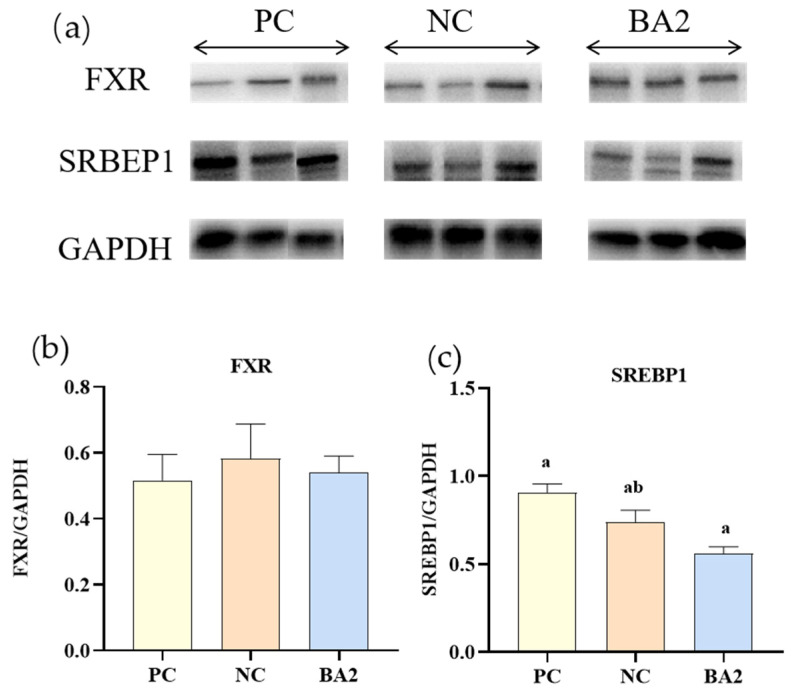
The impact of supplementing low fishmeal diets with bile acids on the protein deposition of *L. vannamei*. (**a**) The western blot analysis of FXR (farnesoid X receptor), SREBP1 (sterol regulatory element binding proteins 1), and GAPDH in the hepatopancreas. (**b**,**c**) The relative quantification of protein levels of FXR and SREBP1 normalized to the GAPDH. Data represent mean ± SD (*n* = 4). Values in the same row with different letters are significantly different (*p* < 0.05) based on Tukey’s multiple-test. PC, 25% fishmeal; NC, 12.5% fishmeal; BA1, NC + 0.03 g kg^−1^ chenodeoxycholic acid; BA2, NC + 0.06 g kg^−1^ chenodeoxycholic acid; BA3, NC + 0.09 g kg^−1^ chenodeoxycholic acid.

**Table 1 animals-13-02109-t001:** Formulation and proximate of experimental diets (% dry matter).

Ingredients	Diets				
	PC	NC	BA1	BA2	BA3
Fishmeal ^1^	25.00	12.50	12.50	12.50	12.50
*Clostridium autoethanogenum* protein ^2^	0.00	10.00	10.00	10.00	10.00
Fish oil	1.50	2.70	2.70	2.70	2.70
Soybean oil	2.50	2.50	2.50	2.50	2.50
Soybean lecithin	1.00	1.00	1.00	1.00	1.00
Soybean meal	25.00	25.00	25.00	25.00	25.00
Peanut meal	10.00	10.00	10.00	10.00	10.00
Shrimp shell meal	5.00	5.00	5.00	5.00	5.00
Beer yeast	3.00	3.00	3.00	3.00	3.00
Wheat flour	23.55	23.55	23.55	23.55	23.55
Vitamin C	0.10	0.10	0.10	0.10	0.10
Choline chloride	0.30	0.30	0.30	0.30	0.30
Vitamin and Mineral Premix ^3^	1.00	1.00	1.00	1.00	1.00
Ethoxyquin	0.05	0.05	0.05	0.05	0.05
Cellulose microcrystalline	1.00	0.80	0.77	0.74	0.71
CaH_2_PO_4_	1.00	2.50	2.50	2.50	2.50
Chenodeoxycholic acid	0.00	0.00	0.03	0.06	0.09
Proximate composition					
Dry matter	89.72	89.84	89.9	89.94	89.86
Crude protein	44.15	43.69	44.10	44.09	43.71
Crude lipid	8.54	8.48	8.66	8.65	8.31
Gross energy (KI/g)	20.04	20.33	20.37	20.35	20.29

PC, 25% fishmeal; NC, 12.5% fishmeal; BA1, NC + 0.03 g kg^−1^ chenodeoxycholic acid; BA2, NC + 0.06 g kg^−1^ chenodeoxycholic acid; BA3, NC + 0.09 g kg^−1^ chenodeoxycholic acid. ^1^ Fishmeal: Peruvian fishmeal, 68.21% crude protein, 9.00% crude lipids, provided by Technology de Alimentos S.A., Callao, Peru. ^2^ CAP, *Clostridium autoethanogenum* protein; 84.20% crude protein, 0.2% crude lipids, provided by Hebei Shoulang New Energy Technology Co. Ltd., Tangshan, China. ^3^ Vitamin and Mineral Premix (kg^−1^ of diet): were provided by Beijing Enhalor International Tech Co., Ltd., Beijing, China.

**Table 2 animals-13-02109-t002:** Primers used for quantitative real-time PCR.

Gene Name	Forward Primers (5′−3′)	Reverse Primers (5′−3′)	GenBank No.
*β-actin*	CCACGAGACCACCTACAAC	AGCGAGGGCAGTGATTTC	AF300705.2
*proPO*	TCCATTCCGTCCGTCTG	GGCTTCGCTCTGGTTAGG	AY723296
*lgbp*	CCATGTCCGGCGGTGGAA	GTCATCGCCCTTCCAGTTG	AY723297
*ppaf*	GAGAAGGAGCTGAACCTGTAC	AGCGCCTGAGTTGTAGTTAG	JX644454.1
*tgl*	ACAAGGTGGATAAGGAAGAG	TAATCAGTAGTTGGCGAAGA	XM_027361886.1
*cpt-1*	CAACTTCTACGGCACTGAT	GTCGGTCCACCAATCTTC	XM_027361886·1
*ampk*	TCAGAGGAGGAGCAGGAAC	CCCGAGGTCTAATAGGCAC	KP272117·1
*srebp*	ACCATTGCCACTCCCCTA	GTTGCGTTTCTCGCCTTI	MG770374.1
*hmgcr*	AGGTGCCCACAAAGACACTC	TGATAGTTCCCCAGCCAGGA	XM_027354586.1
*scp*	TCAGAGGAAATGAACGGGGG	TGGAAGCAGTACACACCCTT	XM_027375905.1
Δ*24sr*	TGCTGATTGTGCTACCGCTT	TGCTGATTGTGCTACCGCTT	XM_027382756.1
*sr*	TGCTTGGACCATTCAAGGGG	ACCCGCATAGTCTCTTGTGC	XM_027383297.1
*7dhcr*	AGACCTGTTACGGCTGTTGAG	GACTGGTCGGGACTCCAAAA	XM_027377095.1
*npc1*	CGAAGGGGAAAAGCCAGAGT	TTGAGGAGGAAGGGAGCGTA	XM_027363410.1

*proPO*: prophenoloxidas, *lgbp*: lipopolysaccharide and beta-1,3-glucan binding protein; *ampk*: AMP-activated protein kinase; *tgl*: triacylglycerol lipase; *cpt-1*: carnitine palmitoyl transferase-1; *srebp*: sterol-regulatory element binding protein; *ppaf*: prophenoloxidase activating factor; *hmgcr*: HMG-CoA reductase membrane form; *scp*: sterol carrier protein 2; Δ*24sr*: Delta (24) sterol reductase; *sr*: Steroid reductase; *7dhcr*: 7-dehydrocholesterol reductase; npc1: *NPC1* intracellular cholesterol transporter 1-like.

**Table 3 animals-13-02109-t003:** Growth performance of *L. vannamei* fed different diets.

Diets					
	PC	NC	BA1	BA2	BA3
IBW (g)	0.25 ± 0.03	0.25 ± 0.03	0.25 ± 0.03	0.25 ± 0.03	0.25 ± 0.03
FBW (g)	5.15 ± 0.06 ^a^	4.79 ± 0.05 ^b^	4.89 ± 0.08 ^b^	4.96 ± 0.10 ^ab^	4.77 ± 0.09 ^b^
WG (%)	1927.18 ± 23.78 ^a^	1787.45 ± 18.08 ^b^	1823.38 ± 29.92 ^b^	1853.51 ± 38.18 ^ab^	1778.39 ± 36.95 ^b^
SGR (%d^−1^)	5.37 ± 0.02 ^a^	5.25 ± 0.02 ^b^	5.28 ± 0.03 ^b^	5.31 ± 0.04 ^ab^	5.24 ± 0.04 ^b^
SR (%)	95.56 ± 3.85	95.56 ± 7.70	100 ± 0.00	95.56 ± 3.85	94.44 ± 6.94
FCR	1.17 ± 0.04	1.17 ± 0.04	1.16 ± 0.01	1.22 ± 0.03	1.22 ± 0.05
FI (g/shrimp)	6.35 ± 0.18	6.14 ± 0.31	5.94 ± 0.12	6.35 ± 0.23	6.16 ± 0.18
PER	1.77 ± 0.02 ^a^	1.67 ± 0.04 ^c^	1.75 ± 0.02 ^ab^	1.72 ± 0.01 ^abc^	1.69 ± 0.02 ^bc^

Data represent mean ± SD (*n* = 4). Values in the same row with different letters are significantly different (*p* < 0.05) based on Tukey’s multiple-test. PC, 25% fishmeal; NC,12.5% fishmeal; BA1, NC + 0.03 g kg^−1^ chenodeoxycholic acid; BA2, NC + 0.06 g kg^−1^ chenodeoxycholic acid; BA3, NC + 0.09 g kg^−1^ chenodeoxycholic acid. IBW, initial body weight; FBW, final body weight; WG, weight gain; SGR, specific growth rate; SR, survival rate; FCR, feed conversion rate; FI, feed intake; PER, protein efficiency rate.

**Table 4 animals-13-02109-t004:** Composition of *L. vannamei* fed different diets.

Diets					
	PC	NC	BA1	BA2	BA3
Moisture	77.46 ± 0.27	77.43 ± 0.37	77.10 ± 0.22	77.74 ± 0.23	77.51 ± 0.35
Crude protein	17.28 ± 0.21 ^ab^	17.01 ± 0.11 ^b^	17.87 ± 0.11 ^a^	17.30 ± 0.16 ^ab^	17.64 ± 0.38 ^a^
Crude lipid	1.85 ± 0.06 ^a^	1.64 ± 0.06 ^b^	1.49 ± 0.07 ^bc^	1.41 ± 0.04 ^c^	1.33 ± 0.08 ^c^

Data represent mean ± SD (*n* = 4). Values in the same row with different letters are significantly different (*p* < 0.05) based on Tukey’s multiple-test. PC, 25% fishmeal; NC, 12.5% fishmeal; BA1, NC + 0.03 g kg^−1^ chenodeoxycholic acid; BA2, NC + 0.06 g kg^−1^ chenodeoxycholic acid; BA3, NC + 0.09 g kg^−1^ chenodeoxycholic acid.

**Table 5 animals-13-02109-t005:** Biochemical indices of hemolymph and hepatopancreas of *L. vannamei* fed different diets.

Diets					
	PC	NC	BA1	BA2	BA3
Haemolymph					
TG (mmol/L)	3.67 ± 0.15	3.51 ± 0.36	3.63 ± 0.09	3.67 ± 0.26	3.57 ± 0.22
T-CHOL (mmol/L)	3.16 ± 0.23 ^a^	1.62 ± 0.16 ^b^	1.50 ± 0.17 ^b^	1.73 ± 0.17 ^b^	1.87 ± 0.24 ^b^
HDL-C(mmol/L)	1.36 ± 0.29 ^abc^	0.81 ± 0.32 ^c^	0.84 ± 0.16 ^bc^	1.59 ± 0.32 ^a^	1.56 ± 0.22 ^ab^
LDL-C(mmol/L)	0.62 ± 0.04 ^a^	0.23 ± 0.03 ^c^	0.19 ± 0.02 ^c^	0.39 ± 0.04 ^b^	0.51 ± 0.09 ^ab^
ALT (U/L)	5.83 ± 1.35 ^b^	11.33 ± 2.24 ^a^	5.65 ± 1.04 ^b^	4.17 ± 1.25 ^b^	3.75 ± 0.08 ^b^
AST (U/L)	21.29 ± 1.85 ^ab^	23.94 ± 1.52 ^a^	18.94 ± 1.95 ^ab^	18.73 ± 3.03 ^ab^	16.87 ± 2.50 ^b^
Hepatopancreas					
T-CHOL (mmol/L)	0.93 ± 0.13 ^a^	0.60 ± 0.05 ^cd^	0.51 ± 0.05 ^d^	0.72 ± 0.05 ^bc^	0.81 ± 0.04 ^ab^
LPS (U/gprot)	1.07 ± 0.28 ^b^	1.52 ± 0.50 ^ab^	2.49 ± 0.54 ^a^	2.04 ± 0.27 ^ab^	2.02 ± 0.31 ^ab^

Data represent mean ± SD (*n* = 4). Values in the same row with different letters are significantly different (*p* < 0.05) based on Tukey’s multiple-test. PC, 25% fishmeal; NC, 12.5% fishmeal; BA1, NC + 0.03 g kg^−1^ chenodeoxycholic acid; BA2, NC + 0.06 g kg^−1^ chenodeoxycholic acid; BA3, NC + 0.09 g kg^−1^ chenodeoxycholic acid. TG, Triglyceride; T-CHOL, Total cholesterol; HDL-C, High-density lipoprotein; LDL-C, Low-density lipoprotein; ALT, Glutathione aminotransferase; AST, Glutathione transaminase; T-CHOL, Total cholesterol; LPS, lipas.

## Data Availability

The corresponding authors can provide data on request to support the findings of this study. It is important to note that the data will not be made public for privacy or ethical reasons.

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
