# Peer review of "Effects of Dietary Chenodeoxycholic Acid Supplementation in a Low Fishmeal Diet Containing Clostridium autoethanogenum Protein on Growth, Lipid and Cholesterol Metabolism, and Hepatopancreas Health of Litopenaeus vannamei"

_animals, 2023, doi:10.3390/ani13132109_

Round 1

Reviewer 1 Report

This manuscript is worthy to publish because it is innovative, practical, and interesting for aquaculturists.

In general, CDCA or chenodiol is produced naturally by intestinal bacteria in the body, and the derivatives of this mineral are secreted in very small amounts by the liver into the gallbladder. There are some minor comments/corrections/concerns as follows:

·         L24, L65, etc.: Please correct “fish meal” to fishmeal throughout the manuscript to be uniform.

·         L42,43: Explain the acronyms fully for the first time.

·         The results part of the abstract section needs to expand more.

·         What was the final conclusion in the abstract section? what was the recommended dose(s) to reach a high productivity?

·         L50: Please correct to: ….in the shrimp fed with BA2

·         L53: what do you mean by slightly damaged? did you use any scale to distinguish the damage stages?

·         L69-71: it is suggested to mention the literature on corn protein concentrate as a partial fishmeal replacer to support the statement.

·         L99: Please, briefly explain the metabolism action of CDCA on the intestinal-hepatic cycle after absorption and finally conjugated form are converted into lithocholic acid.

·         Add the gross energy for the experimental diets in Table 1.

·         If there is information on the proximate composition of the fishmeal and CAP, add them in the footnote of Table 1.

·         In Table 1, the authors used Peanut meal to supply which nutritional requirement(s) for whiteleg shrimp?

·         L152, L153: Please change to … alanine aminotransferase (ALT), and aspartate aminotransferase (AST) in the hemolymph….

·         Why the authors only used one housekeeping gene (beta- actin) to normalize the gene expression Ct?

·         In Table 3, please add the survival rate of the experimental groups. Because it is practical in aquaculture and interesting to know for readers.

·         Plz explain the experimental groups’ abbreviations in the captions or footnotes of Tables and Figs. In All tables, the authors should also provide the description of the abbreviations, statistical analysis, P-value, and number of replicates as n=?.

·         Please do not forget to italic the scientific names (e.g. L243).

·         The discussion section was strong and informative.

·         L312: why the higher dose of CDCA had a negative impact? In which probable mechanistic view(s)? Supplement this part with reason(s).

·         L336-340: Finally, it is a good sign to reduce T-CHOL and HDL-C or increase them in the shrimp production?

·         L402: Please illustrate this issue in Fig. 3.

·         No discussion was given for the western blot analysis.

Reviewer 2 Report

The authors Shia et al investigated henodeoxycholic acid supplementation in a low fishmeal diet containing Clostridium autoethanogenum protein on Litopenaeus vannamei. I would suggest the authors to consider some points discussed below and revise the manuscript accordingly.

First, the authors has to find suitable representation to write the % of fishmeal because the positive control is having 25% fishmeal, and only 12.5% was used for the experimental diets. When authors say 50% fishmeal then it is misleading. So authors has to modify the lines that would convey the exact amount of fishmeal so the readers would not be confused especially in the simple summary and abstract.

Line 36-37: five diets supplemented with CDCA. But actually without positive control it is only 4 diets so authors rephrase the line.

Line 61-63 The same line has been repeated twice. Please check

Line 77: “More and more studies….” But there were citation. Please include some citations.

Line 86-91: rephrase the sentence because you said 30% of CAP does not adversely affect the growth but other studies showed that 40% does not affect. Why you have added “But there are other studies” if you say but as a conjunction then the following sentence should be contradictory. Or say “similarly” So rephrase the line.

Line 117: similar to the previous comment check the number of diets. Not all the five diets containing 12.5% fishmeal. Check the line and modify it.

 Line 118: Fishmeal was substituted with 50% CAP but you added 10.00 rather than adding 12.50. In addition you also increased the fish oil content in the experimental feed. Please indicate that whether you changed the formulation to be isoproteinous diets.

Table 1: please give the units for the diet formulations.

Line 129: authors mentioned each group consisted quadruplicates but the results were given in triplicates why (e.g. see the footnotes of table 3).

Figure 1 & 2: Authors have to change the colour of the columns of BA1, BA2 & BA3. All these three shows similar colours and it is hard to distinguish.

Line 317: Check the spelling of “addition” likewise I have seen some spelling mistakes in the manuscript. Authors have to check the spelling thoroughly in the manuscript.

Round 2

Reviewer 2 Report

The authors Shia et al made some significant corrections in the revised manuscript but still it needs minor revision.

Instead of saying 50% replacement authors used half the but it is not scientifically written. I suggest to use partial for the line no.27 please change it as given below.

 “The results revealed that the partial replacement of fishmeal with Clostridium autoethanogenum protein”

For line 36: change to “fishmeal was partially replaced with Clostridium autoethanogenum protein in four diets”

The authors have not understood the comment which I said last time. Check line 63-65. The first sentence and sentence is the same and repeated again. Check carefully and remove the second sentence.

Line 231: remove “of three replicates” just write mean ± SD (n = 4). If you say three replicates and indicating (n=4) is contradictory. Similarly remove the word “of three replicates” in the other mentioned lines.

Line 329: change to “The study revealed that the partial replacement of fishmeal with CAP”

Line 370: remove “half the” simply write “CAP replacement of fishmeal”

Line 425: remove “half the”

Line 449: remove “half”
